# The racialization of pit bulls: What dogs can teach us about racial politics

**Michael Tesler** [ID]*, **Mary McThomas**

University of California, Irvine, California, United States of America

* mtesler@uci.edu

**Data Availability Statement:** All relevant data are within the manuscript and its Supporting information files.

**Funding:** The authors received no specific funding for this work.

## Abstract

Many have argued that discrimination against pit bulls is rooted in the breed's association with Black owners and culture. We theoretically and empirically interrogate that argument in a variety of ways and uncover striking similarities between the racialization of pit bulls and other racialized issues (e.g., poverty and crime) in public opinion and policy implementation. After detailing the reasons to expect pit bulls to be racialized as Black despite dog ownership in the U.S. generally being raced as white, the article shows: (1) Most Americans associate pit bulls with Black people. (2) Anti-Black attitudes, in general, are significant, independent, predictors of both anti-pit views and of preferring other breeds over them; (3) stereotypes of Black men as violent, in particular, are significant, independent, predictors of both anti-pit views and of preferring other breeds over them. (4) Implicit racialization through a national survey experiment further eroded support for legalizing pits, with the treatment effect significantly conditioned by respondent's race. And (5) state-level racial prejudice is a significant negative predictor of enacting legislation to preempt breed-specific bans. We conclude with our findings' broader insights into the nature of U.S. racial politics. Michael Tesler, mtesler@uci.edu, corresponding author, is Professor of Political Science at UC Irvine; Mary McThomas, mary.mcthomas@uci.edu, is Associate Professor of Political Science at UC Irvine. An earlier version of this paper was presented at the American Political Science Association's annual meeting. We thank Maneesh Arora, Rachel Bernhard, Nathan Chan, Louis Pickett, David Sears, DeSipio, Adam Duberstein, Jane Junn, Claire Kim, Jessica Manforti, J. Scott Matthews, Justin.

## Introduction

> Separately, the young Black man and the pit bull make people cross to the other side of the street; together, they are a picture of unmitigated threat."
>
> -Claire Jean Kim, *Dangerous Crossings*. pg. 272. [1]

Racialization occurs when an issue, person, policy, or even a specific dog breed becomes infused with racial connotations [2]. Crime and poverty, for example, are thought to be

**Competing interests:** The authors have declared
that no competing interests exist.

racialized via their connections to African American perpetrators and welfare recipients [2–6]. In both instances there is an empirical link between race and the issue, as African Americans are disproportionately more likely to be incarcerated and receive welfare than white people. But those associations are then framed in an especially negative light by the news media, which in turn helps erode racially prejudiced whites' support for welfare and compassionate criminal justice policies. The implementation of these racialized policies across the states then reflects their underlying racialization, as the most racially prejudiced states have less generous welfare benefits and more punitive crime policies (see discussion below).

Many have suggested that pit bulls are similarly racialized. Several scholars and journalists have even argued that the widespread and well-documented societal prejudice and discrimination against pit bulls [7,8] is at least partially rooted in the dog breed's association with Black men and African American culture. A 2016 op-ed in the *Washington Post*, for example, contended that breed-specific legislation (BSL) banning pit bulls are likely "proxies by which uneasy majorities can register their suspicions about the race, class and ethnicity of the people who own those dogs" [9]. Katja Guenther [10, pg 155–156] similarly states, "pit bulls are now 'raced Black,' and, like Black men, they are consequently subjected to discriminatory policies and practices based on fear of the risk they purportedly pose to whites, to public safety, and to the social order." After documenting the racialization of pit bulls, Claire Kim [1, pg 273] concludes, "Pit bulls are dying for being Black." And Bénédicte Boisseron [11, pg. 152] argues in her book, *Afro-Dog*, "When one becomes aware of the racialization of the pit bull through putative black ownership, pit bull bans across America take on the appearance of a modern version of the plantation-era ban [on Black people owning canines]."

Those arguments are compelling and plausible. After all, the pages that follow detail the long line of academic research showing that the news media, the white public, and politicians are often less sympathetic to issues and policies after they're racialized via negative associations with African Americans. But there have been few quantitative analyses of how race and racial attitudes affect public opinion and public policy toward pit bulls. Moreover, Thompson, Pickett, and Intravia's [12] recent empirical study found that neither experimentally linking pit bull owners with African Americans via racial imagery, nor harboring stereotypes of pit bull owners as disproportionately Black, were associated with support for banning the breed among college students.

Thompson et al., however, conclude that study by noting the need to replicate their findings with representative samples. They specifically discuss how the young age of their college sample could affect the results if attitudes toward pit bulls have improved since Michael Vick's 2007 dog-fighting controversy. This is a particularly prescient point. Consistent with the unusually low levels of support for breed-specific bans in Thompson et al.'s data (1.74 on a 1–5 scale), we note multiple times throughout the manuscript that age is one of the strongest predictors of public opinion about pit bulls presumably because American youths have been socialized amidst increasingly positive images of the breed on social media. Younger and college-educated Americans score significantly lower in racial prejudice (see discussion below), as well, which could help further explain why the link between pit bulls and African Americans did not increase support for breed-specific legislation among college students.

So, there are still lots of unanswered questions about the racialization of pit bulls. Does the public, for example, really associate pit bulls with African Americans? If so, has that association eroded support for these dogs, especially among racially prejudiced whites? And are racial prejudice and the politics of race linked to legislation banning these dogs from certain neighborhoods? This article explains the causes and consequences of racializing pit bulls by providing detailed empirical answers to these questions. In doing so, we reveal some striking similarities between the racial politics surrounding pit bulls and the racialization of ostensibly

non-racial issues, such as poverty and crime, in both public opinion and in policy implementation across the states. Our analyses of pit bulls, in fact, offer some broader insights into the nature of racial politics in the United States that we discuss in the conclusion.

## The racialization of dog ownership and pit bulls

Dog ownership has historically been racialized as white in American society [11,13]. The top panel of Fig 1 shows the modern-day manifestation of that racialization—the large racial divide between Black and white Americans in dog ownership. Whites were roughly twice as likely to own a dog as African Americans were in both a 2005 Pew Poll and in the 2008 National Annenberg Election Survey, with less than one-quarter of African Americans having a pet dog in both surveys. It's hardly surprising, then, that white people interacted more frequently with dogs and rated them more favorably than African Americans did in six combined national surveys (pooled N > 6,000) we fielded from November 2018 to August 2021 through Lucid—a relatively new opt-in online polling firm whose demographics and experimental treatment effects track well with findings from U.S. probability samples [14,15].

The racial dog-divide is rooted in a variety of factors, including socioeconomic inequality between the races [16] and the legacy of plantation-era laws banning Black people from owning canines [11]. Since respondents who had a pet dog during childhood were nearly twice as likely in our Lucid surveys to own a dog in adulthood as those who didn't (57% to 30% respectively), the effects of these antebellum restrictions have likely been passed down from generation to generation. Perhaps an even bigger factor, though, is "the recurrent history, on both sides of the Atlantic, of canine weaponry used against the oppressed." [11, pg. 153]." From Bloodhounds tracking and attacking fugitive slaves, to German Shepherds mauling civil rights protestors in the 1960s, to the Ferguson Police Department's more recent practice "of deploying canines to bite individuals when the articulated facts do not justify this significant use of force" [17] white authorities have used dogs to terrorize and control Black people.

Despite that history, however, the bottom panel of Fig 1 shows that African Americans are more likely than white people to regularly interact with pit bulls—a modest but highly significant six percentage-point difference. You can see in the display that African Americans are also significantly more likely than whites to say pits are their favorite breed and less likely to rate the breed as their least favorite type of dog. There were not significant differences between the races in how favorably they rated pit bulls; but after controlling for the fact that white people have more favorable views about dogs than Black people do, and that attitudes about dogs in general strongly predict attitudes towards pit bulls (see Table 1), African Americans were significantly more likely than whites to rate pit bulls very favorably (40% to 32% respectively, p < .001).

Pit bulls gained popularity in African American communities during the late twentieth century as protectors who "afforded security and status to men who feared violence from police and peers" [10, pg 154; see also 1, 18–20]. That link was further solidified by the pit bull's racialized place in media and popular culture. Bronwen Dickey [18, pg 220], for example, describes pit bulls as hip hop's "unofficial mascot," with several of the genre's biggest stars owning and/or appearing in music videos with the breed. And negative associations between pit bulls and African Americans have been reinforced by media and pop-culture portraits of African Americans' involvement in illegal dogfighting operations [1,10,19].

The upshot is that the public predominantly thinks of pit bulls as a Black-owned dog breed —much the way that they disproportionately associate welfare and violent crime with African Americans [3,21]. In three of our Lucid surveys, which were fielded in June 2020, July 2020, and August 2021, we asked our respondents, "If you had to guess, do you think that white

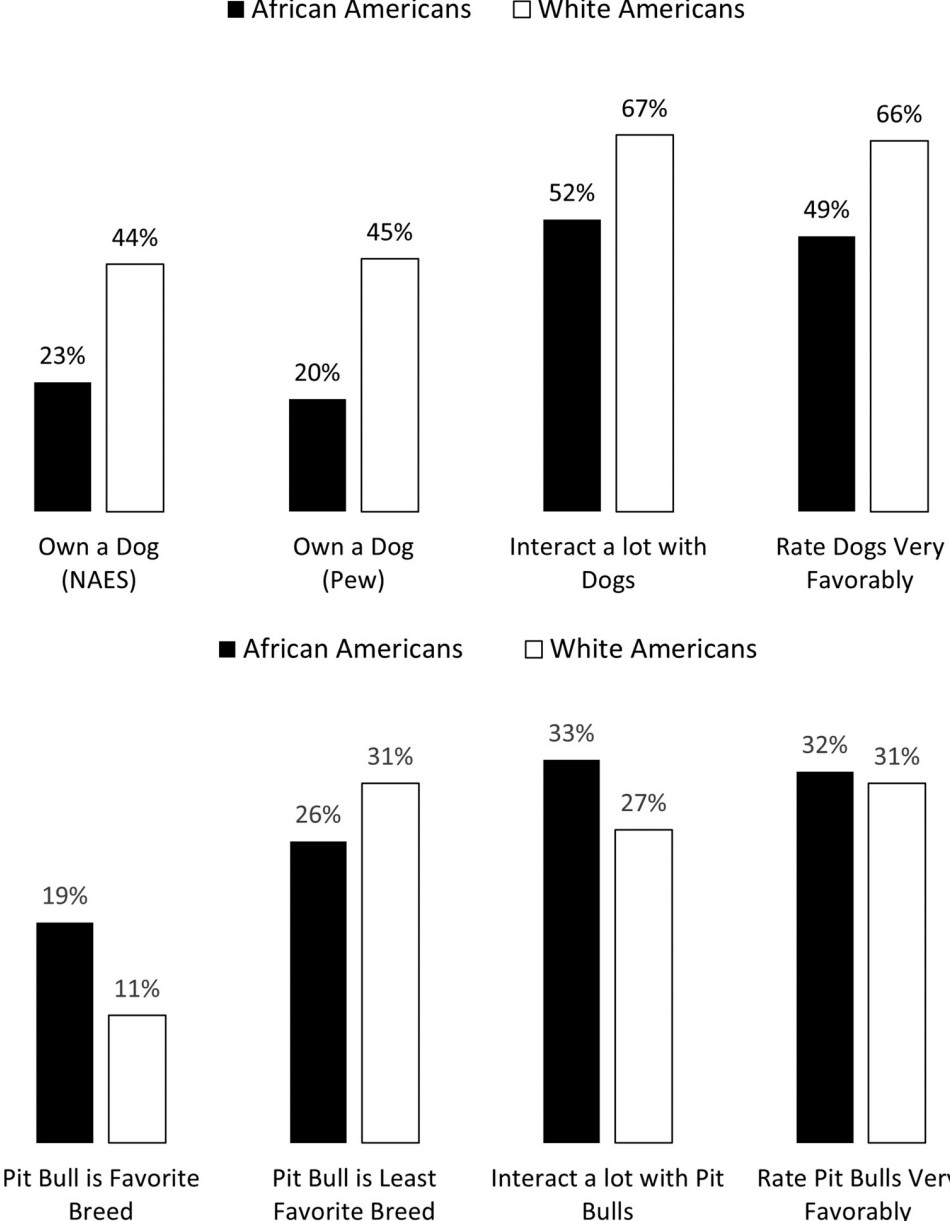

**Fig 1. The racial divide over dogs and pit bulls.** Sources: 2008 National Annenberg Election Survey; Pew Social Trends Poll, Oct-Nov 2005; Pooled Lucid Surveys, 2018–2021.

people or black people are more likely to own the following dogs?" Fig 2 shows that white people are perceived as much more likely to own such popular dog breeds as Golden Retrievers, Collies, Labradors, and Dalmatians, which is again consistent with the idea of dog ownership being generally raced as white. But most of our respondents thought African Americans are more likely than whites to own pit bulls and Rottweilers.

It's hardly a coincidence, either, that the two dog breeds stereotyped as Black-owned are the two breeds that evoke the most fear from the public. Over 40 percent of respondents in a 2018 Lucid survey we conducted, for example, said that "scary" described pit bulls and rottweilers

**Table 1. (OLS) predictors of white Americans opinions of pit bulls and other dog breeds.**

| | Net-Fav Pit Bulls | Net-Fav 9-Dog Scale | Difference: Pit minus Dog Scale | Net-Fav Pit Bulls | Net-Fav 9-Dog Scale | Difference: Pit minus Dog Scale |
|---|---|---|---|---|---|---|
| Blacks Favorability | .371*** | .184*** | .180*** | | | |
| | (.049) | (.019) | (.046) | | | |
| Whites Favorability | -.026 | .078*** | -.103* | | | |
| | (.056) | (.023) | (.051) | | | |
| Dogs Favorability | 1.08*** | 1.12*** | -.043 | 1.26*** | 1.19*** | .076 |
| | (.057) | (.022) | (.054) | (.129) | (.054) | (.118) |
| Party Identification | -.024 | .013 | -.039 | -.016 | .013 | -.029 |
| | (.034) | (.013) | (.032) | (.079) | (.033) | (.072) |
| Actual Age | -.016*** | .001*** | -.017*** | -.010*** | .002* | -.012*** |
| | (.001) | (.000) | (.001) | (.002) | (.001) | (.002) |
| Education | -.066 | -.076*** | .008 | -.136 | -.076 | -.060 |
| | (.051) | (.020) | (.048) | (.108) | (.045) | (.099) |
| Male | .049 | .012 | .038 | -.176** | -.069* | -.106 |
| | (.026) | (.010) | (.023) | (.064) | (.027) | (.058) |
| Violent: Black Men | | | | -.507** | -.086 | -.421** |
| | | | | (.173) | (.072) | (.158) |
| Violent: Black-Wom | | | | .119 | .032 | .087 |
| | | | | (.179) | (.075) | (.163) |
| Violent: White Men | | | | .127 | .041 | .086 |
| | | | | (.159) | (.066) | (.145) |
| Violent: White-Wom | | | | .294 | -.092 | .387* |
| | | | | (.179) | (.075) | (.163) |
| Violent: Muslim Men | | | | -.002 | .058 | -.060 |
| | | | | (.156) | (.065) | (.143) |
| Violent:Muslim-Wom | | | | .213 | .088 | .124 |
| | | | | (.174) | (.073) | (.159) |
| Constant | -.161* | -.543*** | .383*** | -.213 | -.410*** | .197 |
| | (.076) | (.030) | (.071) | (.167) | (.070) | (.153) |
| $R^2$ | .166 | .407 | .119 | .170 | .389 | .125 |
| Observations | 4545 | 4536 | 4535 | 860 | 860 | 860 |

Significance codes:

*$p < .05$,

**$p < .01$,

***$p < .001$.

Note: Dependent variables range from -1 (rate unfavorably) to 1 (rate favorably). All explanatory variables except actual age are coded from 0 to 1, with 1 representing the highest or most conservative. value. *Source*: Pooled Lucid Surveys 2018–2021; Lucid Survey, August 2021 (Right-hand columns); white respondents only.

"extremely" or "very well" (46 and 41 percent, respectively). But only about 10 percent said the same thing about golden retrievers, collies, Dalmatians and Labradors. These four breeds were all rated at least 15 percentage points more favorably than rottweilers and pit bulls in our surveys as well [22]. To be sure, there are plenty of other reasons why people may find these dogs scary, such as their large muscular builds, history of being bred for fighting purposes, and rare but sensationalized involvement in fatal attacks. Yet, the theoretical background and empirical evidence provided in the following sections indicate the breed's associations with African Americans has played an important and independent part in prejudice and discrimination against pit bulls.

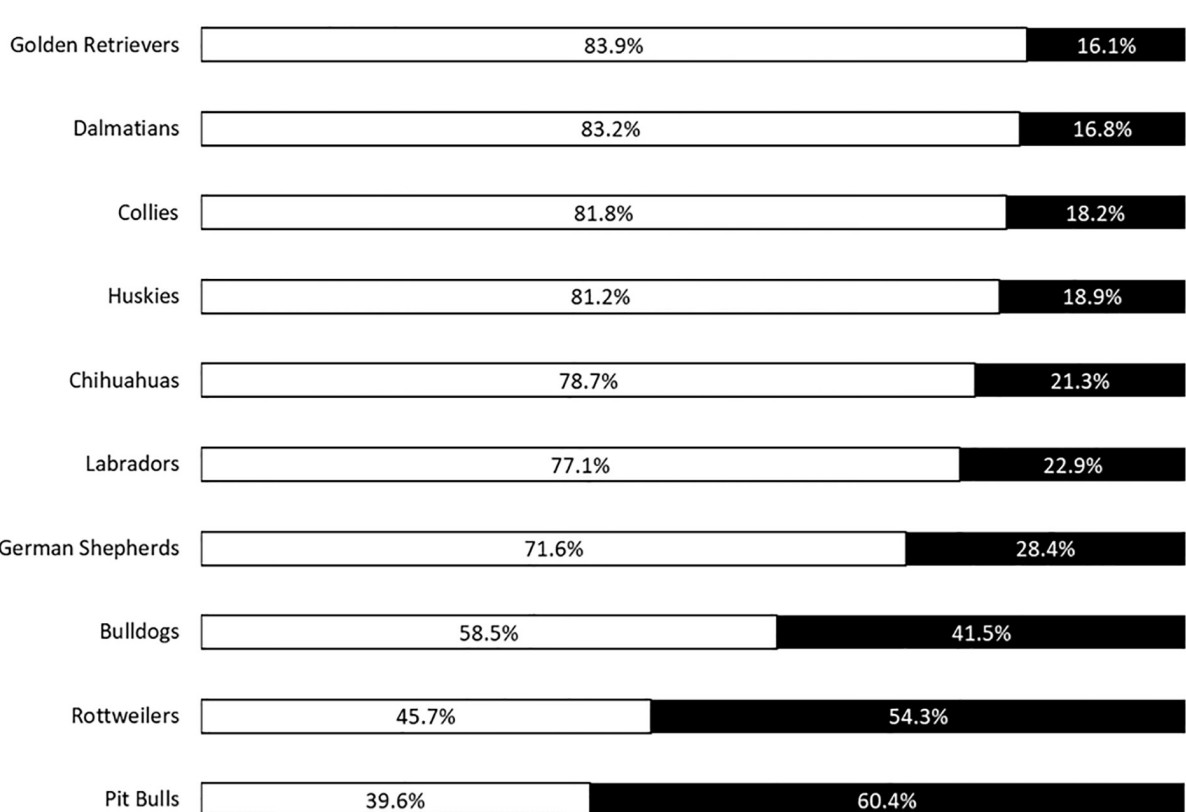

**Fig 2. Perceptions of whether black or white people are more likely to own certain dog breeds.** Note: Question asked respondents: "If you had to guess, do you think Black people or white people are more likely to own the following dogs?" *Source*: Pooled Lucid Surveys, June-July 2020, August 2021.

## Theoretical background and empirical expectations

A large body of research explains the causes and consequence of racialization in American politics. This process of racialization, whereby certain issues, policies, and people are inextricably associated with specific racial groups, occurs in a variety of ways. Some issues like affirmative action and reparations for African Americans automatically evoke race because there's such a clear link between the policy and the groups who benefit from them [23]. Barack Obama's "embodiment of race" as the country's first African American president made it similarly easy to project hopes and fears about race onto his presidency [24,25]. So much so, in fact, that issues like health care were racialized simply through their connection to his presidency [25–27]. Most importantly for our purposes, this "spillover of racialization," extended all the way into feelings about the Obamas' dog, Bo [25].

The media often plays a critical role in the racialization process as well. Prior research suggests that the racialization of issues stems in large part from mass communications, which strengthen their association with specific racial groups [2,3,28]. Those associations usually begin with some empirical basis, such as higher incarceration rates among African Americans or white Americans disproportionately dying of opioid overdoses. But the news media often frames those issues differently depending on whether they have a Black or white face attached

to them. Many astute analysts, for example, have documented how different the media's sympathetic coverage of rural white opioid users has been from their negative stories about the crack epidemic in Black communities during the 1980s [29–33]. As Raychaudhuri, Mendelberg, and McDonough [34, pg. 168] conclude, "drugs associated with racial minorities are framed with negatively-valenced topics such as crime, while drugs associated with Whites are characterized with positively-valenced topics such as community and family."

Those different portraits of Black drug use fit well with prior social science research on race, media, and attribution error [3,35–37]. The upshot of those cognitive biases is that when white Americans struggle, their troubles are usually attributed to situational forces like overprescribed painkillers. But when non-whites struggle, their plight is more often attributed to negative dispositional traits, such as the group's supposed poor work ethic and lax moral values. Martin Gilens [3], for instance, famously found that images of Black poverty in newsmagazines reinforced negative racial stereotypes of African Americans as "the underserving poor," while impoverished whites were portrayed more favorably as victims of economic conditions beyond their control. The news media tends to portray Black criminality more menacingly, too, with Robert Entman and Andrew Rojecki [38, pg. 82] finding "a tendency for Blacks accused of crimes to be portrayed as individuals less than whites—that is, to be lumped together without distinct identities and laden with negative associations."

We suspect that the media might also portray pit bulls especially unfavorably when they are linked up with African Americans. Qualitative research certainly suggests that they do. We noted earlier, for example, that scholars have documented the ways in which negative associations between pit bulls and Black Americans are propagated by media and pop-culture portraits of African Americans' involvement in illegal dogfighting operations [1,10,19]. Dickey similarly chronicles the "dark-skinned" imagery and racial fears that were so prominent in the late twentieth century news coverage of "the pit bull panic" [18]. And our subsequent research provides some suggestive quantitative evidence on the heightened negativity of news stories about pit bulls when they're associated with African Americans [39]. Drawing on automated sentiment analyses from Gary King's analytics platform, Crimson Hexagon, we found that the net sentiment (positive minus negative) of headline news stories about pit bulls, which explicitly mentioned African Americans, was significantly lower ($p < .001$) than the already high levels of negativity in all news coverage of the breed (-68 to -48 respectively).

Those negative associations between African Americans and pit bulls in media and popular culture could certainly affect public opinion. Indeed, a large volume of political science research shows that racialized news coverage can heighten the association between racial attitudes and white Americans' policy preferences [40,41]. The emergence of news coverage linking welfare benefits with "undeserving Blacks" in the 1960s and 1970s eroded support for this policy, especially among racially prejudiced whites [2,3,6,42]. Meanwhile, news coverage connecting Social Security to hardworking white recipients who are getting their just rewards helped make this policy popular, especially among ethnocentric whites who rate their own group higher than racial and ethnic minorities [2,43]. Social scientists have similarly argued that media coverage, which exaggerates Black violence, have helped make racially resentful whites' more supportive of the death penalty and other punitive criminal justice policies [44–46]. Chiricos et al. [21] relatedly found that white Americans significantly overestimated the share of violent crime committed by African Americans and that those misperceptions were linked to support for harsher criminal justice penalties.

Based on those studies, our first formal hypothesis, H1, posits that *anti-Black attitudes are a significant, independent, predictor of negative opinions about pit bulls*. Relatedly, we also expect that racial prejudice will be more strongly associated with public opinion about pit bulls than

it is with other dog breeds. Or, stated more formally, H2 *expects anti-Black attitudes to significantly predict rating pit bulls less favorably than other dog breeds.*

Issues like welfare and crime aren't just racialized, though. They also evoke intersectional stereotypes of African American women and Black men respectively [47]. Derogatory portraits of Black women as promiscuous and irresponsible "welfare queens," for example, have long been weaponized against anti-poverty policies. So, it's not surprising that there's an especially strong empirical link between opposition to welfare and stereotypes of Black women as sexually irresponsible [6,48]. Nor is it surprising that widespread societal stereotypes of Black men as violent factor heavily into perceptions of crime and public opinion about the criminal justice system. [45,47,49,50].

These same exaggerated fears of Black men might also be implicated in opposition to pit bulls. After all, Guenther [10, pg 154–55] describes pit bulls as "synonymous with Black masculinity," stating, "This link between Black masculinity and pit bulls was and still is captured and reinforced through the image of pit bulls as companions to Black male 'thugs' depicted in hip-hop and rap media and in the mainstream media, and of pit bulls as part of illegal dog-fighting operations involving Black men in both poor urban neighborhoods and the rural south." Our next formal hypothesis, therefore, is that *stereotypes of Black men as violent will be a significant independent predictor rating pit bulls unfavorably even after controlling for stereotypes of Black women and other groups as violent (H3).*

As important as observational studies have been in showing how racialized media coverage can affect public opinion, social science experiments provide even stronger evidence. Several experiments, which randomly assigned subjects to receive *implicit racialized messages* (e.g., racial images and/or race-coded language that does not explicitly reference a particular racial group) about specific issues, have affected white Americans' opinion about crime, welfare, drug treatment, gun control, government spending, education programs, Social Security, the minimum wage, the Iraq War, and the coronavirus pandemic [2,3,34,43,45,51–55]. Based on those studies, H4 posits that *implicitly linking pit bulls with African Americans will significantly increase white Americans' support for banning the breed.*

Finally, the racialization of issues like poverty and crime often has important policy consequences. For, as Beth Reingold and Adriene Smith [56, pg. 131] surmised, "State lawmakers have responded to or internalized the racial stereotypes, resentments, and fears that shape judgments of welfare recipients and drive the call for less generous, get-tough welfare policy among whites." In keeping with that contention, states that score higher in measures of white racial prejudice were less likely to expand Medicaid coverage under the Affordable Care Act and have fewer social welfare benefits, on average, than states whose citizens have more progressive views about race [57–59]. States with large Black populations were also less likely to expand Medicaid and have more rigid rules and regulations governing eligibility and work requirements for welfare benefits [6,60–62]. Likewise, states with larger Black populations and more racially prejudiced constituents tend to have more punitive criminal justice policies than other states [63–66]. Our final hypothesis, then, is that *states that score high in racial prejudice will be significantly less likely to enact legislation preempting municipal pit bull bans (H5).*

## Data and methods

This article employs a wide variety of data, measures, and statistical analyses to formally interrogate those hypotheses. To test the suspected association between white racial prejudice and public opinion about pit bulls formally posited in hypotheses H1-H3 we commissioned seven national surveys from 2018 to 2021, each of which sampled at least 1,000 Americans. As we mentioned earlier, six of the surveys were fielded through Lucid between November 2018 and

August 2021 (IRB Exempt Approval HS#2017–3811; see page 3 of the appendix for the informed consent message at the beginning of each survey). While the firm's demographics and experimental treatment effects track well with findings from U.S. probability samples [14,15], Lucid is a relatively new online survey platform. So, we also replicated our findings with data from a more established polling firm, YouGov, sampling 1,000 respondents as part our team's module in the 2018 Cooperative Congressional Election Survey (CCES).

The CCES and Lucid surveys both contained the same four questions about pit bulls that previously appeared in a July 2014 YouGov/HuffPost Poll [67]. Those questions asked: (1) if it should be legal or illegal to own a pit bull; (2) if pit bulls are naturally more aggressive than other breeds; (3) if it is safe or too dangerous for pit bulls to live in residential neighborhoods (asked in two of the six Lucid surveys); and (4) if the respondent would personally consider adopting a pit bull. In addition to testing the association between racial attitudes and those four dependent variables, all six of our Lucid surveys asked how favorably respondents rated many of the most popular dog breeds—pit bulls, Labradors, Golden Retrievers, German Shepherds, Collies, Huskies, Bulldogs, Dalmatians, Chihuahuas as well as dogs in general (these dog favorability questions were not included in our CCES team survey). Those items are then used to test H2's expectation that anti-Black attitudes are significant predictors of rating pit bulls less favorably than other dog breeds.

The Lucid and CCES surveys also contained two blatant measures of racial prejudice. The first measure uses five-category favorable/unfavorable ratings of African Americans. The second measure is a two-item scale of old-fashioned racism (OFR), which taps into aversion to interracial intimacy and has been validated in prior research [68]. These two items asked respondent how strongly they agreed or disagreed with the following statements (1) "I prefer that my close relatives marry spouses of their same race," (2) "I think it's alright for Blacks and whites to date each other." Finally, we included a question about how well the term "violent" described various racial/gender groups to test H3's contention that exaggerated stereotypes of Black men as violent are significant predictors of negative opinions about pit bulls.

We chose these measures of racial prejudice for a couple of reasons. Most importantly, anti-Black affect, OFR, and anti-Black stereotypes are all only weakly associated with other sociopolitical attitudes [25,69,70], thereby minimizing the risks of spurious correlations with anti-pit attitudes. But we also deployed them because this study is primarily interested in how racial prejudice erodes white support for pit bulls; and our prior research shows that these blatant measures of racial prejudice are unable to adequately identify racially sympathetic whites who may be more supportive of issues linked to African Americans [25,71].

We then augment our observational findings from the CCES and Lucid data with an original survey experiment to test H4's causal claim that implicitly associating pit bulls with African Americans further erodes white support for the breed. The experiment was embedded into three of the Lucid surveys that we fielded in June 2020, July 2020 and August 2021 (pooled N = 3196). Our experimental design followed several prior studies, which all indirectly associated African Americans with specific policies through the racially evocative term "inner-city" [2,28,43].

Lastly, we test H5's contention that states scoring high in racial prejudice will be significantly less likely to enact legislation preempting municipal pit bull bans with Fix and Mitchell's [72] data on the twenty states who passed legislation preventing local governments from banning and regulating dogs solely based on breed (aka, pit bull protection laws) between 1989 and 2016. We then examined the relationship between passing pit bull protection laws and two different measures of state-level racial prejudice, both of which have been used and validated in prior research [73,74]. The first measure calculates each state's level of white opposition to interracial dating from the Pew Values Survey cumulative file—a repeated cross-sectional

survey that interviewed over 30,000 respondents from 1987 to 2012. The second follows Stephens-Davidowitz's [74] approach and measures state-level prejudice with relative rates of racist Google searches for the "N-word". The two measures are highly correlated with one another (r = .78), bolstering our confidence that both are tapping into state-level prejudice. We are also reassured by the fact that both measures of racial prejudice were more strongly associated with state-level opposition to Barack Obama in 2008 than John Kerry in the 2004 presidential election [73,74].

## Racial prejudice and public opinion

Our first hypothesis posited that anti-Black attitudes should be a significant, independent, predictor of negative opinions about pit bulls. Consistent with H1's expectations, unfavorable views of African Americans and OFR were significantly correlated with our four questions about pit bulls in both the CCES and Lucid data (see Table A1 in S1 Appendix).

But it's important to account for other variables to test H1's contention that racial prejudice is an *independent predictor* of public opinion about pit bulls. Our findings are robust to every possible specification we tried, so we selected our final model based on several factors. We followed Winter [2] and included favorability ratings of whites alongside favorability of African Americans to assess the relative influence of each variable while holding the other one constant. Partisanship is in the model to both account for its growing association with racial attitudes [25,75] and to determine the degree to which public opinion about pit bulls is rooted in party polarization. Our earlier discussion of pit bulls and masculinity prompted us to control for sex. We included education to control for its well-documented association with racial attitudes (correlation with OFR = -.17), and to assess how socioeconomic status is linked to public opinion about the breed. Finally, and most importantly, we included age to account for the fact that young people score significantly lower in racial prejudice (age is correlated with ORF at r = .28) and have much more favorable opinions of pit bulls than older Americans. Like most of the research we cited in the previous section on how racial attitudes are associated with public opinion about such issues as welfare, crime, drug addiction, health care, government spending and affirmative action, our analyses in this section are limited to whites. Including other races/ethnicities in the analysis, however, does not alter the substantive or statistical significance of the findings (see Table A4 in S1 Appendix).

Even after accounting for all those factors, Fig 3 shows that white Americans' favorability ratings of African Americans were consistent predictors of support for pit bulls in both surveys. Across the eight dependent variables in the Lucid and CCES surveys, whites who rated African Americans very favorably were considerably more supportive of the breed. The display shows that moving from having a very unfavorable impression of African Americans to a very favorable impression increased sympathetic responses to those items from about 20 to 40 percentage points after controlling for favorability ratings of whites, partisanship, education, sex, and age. All eight relationships were highly significant, too, providing strong support for H1's contention that anti-Black attitudes are correlated with unfavorable opinions of pit bulls. Moreover, the results in Fig 3 replicate across our other measure of overt racial prejudice— old-fashioned racism (see Table A3 in S1 Appendix).

The relationships in Fig 3, however, cannot tell us whether anti-Black attitudes are equally implicated in public opinion about all dog breeds. That seems unlikely in light of our earlier discussion of dog ownership being raced as white, but it's still important to interrogate H2's expectation that anti-Black attitudes are a significant predictor of rating pit bulls less favorably than other dog breeds. As noted above, we test this hypothesis with favorability ratings of pit bulls, Labradors, Golden Retrievers, German Shepherds, Collies, Huskies, Bulldogs,

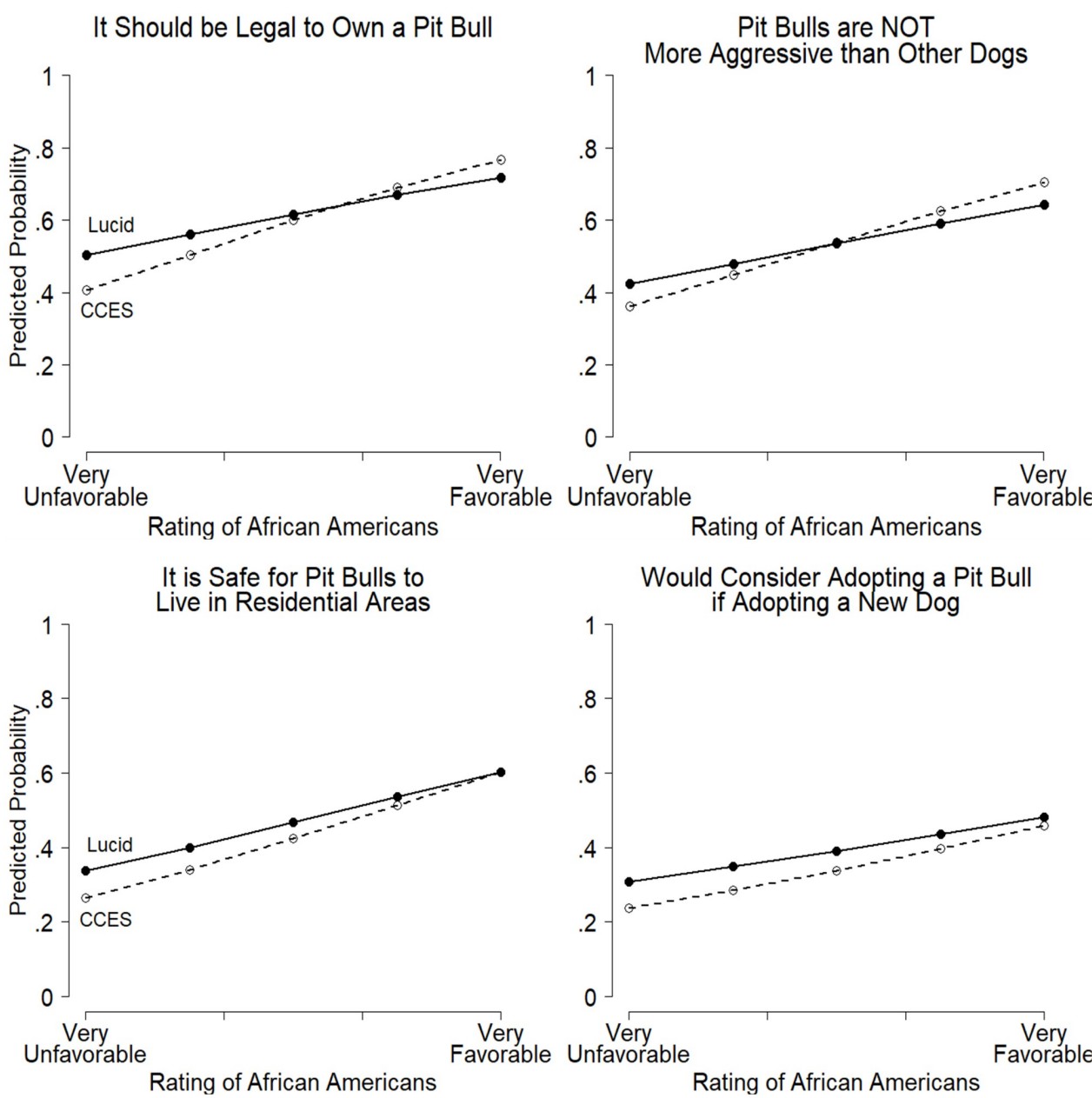

**Fig 3. Favorability ratings of African Americans predict white Americans' support for pit bulls.** Note: Predicted probabilities are based on the logistic regression coefficients in Table A2 in S1 Appendix. Predicted probabilities were calculated by setting favorability rating of whites, partisanship, age, education level and gender to the mean white respondent. Sources: 2018 CCES Team Module; 2018–2021 Pooled Lucid Surveys; white respondents only.

Dalmatians, Chihuahuas and dogs in general—that were included in all six Lucid surveys. We then scaled the non-pit bull dogs into a nine-item additive index of net dog favorability (Chronbach's alpha = .82).

The first three columns of Table 1 regress our net dog favorability scale, net favorability ratings of pit bulls, and the difference between the two, on the same predictors used in Fig 2. We also control for favorability ratings of dogs in general, which was not asked in our CCES

survey, to account for the fact that dog lovers also have much more positive views of pit bulls. The coefficients in Table 1 reveal that unfavorable impressions of African Americans, favorable views of whites, and older age were all significant predictors of rating pit bulls more negatively than other dog breeds. The results for favorable views of whites are consistent with the idea of dog ownership being generally raced as white in the U.S. and echo Winter's [2] argument that "the racialization of Social Security turns on white Americans' feelings about their own racial group because the policy is linked to white beneficiaries." Meanwhile, the coefficients on Black favorability in Table 1 confirm H2's suggestion that racial prejudice is an important predictor of pit bulls being rated less favorably than other dog breeds. Indeed, white Americans with a very unfavorable impression of African Americans rated other dog breeds much more favorably than pit bulls (+53 to -5 respectively) after holding the other variables in Table 1 at their means.

Finally, the last three columns of Table 1 test H3's contention that exaggerated stereotypes of Black men as violent are significant predictors of public opinion about pit bulls. The fourth column shows that out of the six groups evaluated, only stereotypes of Black men as violent emerged as a significant predictor of holding less favorable views of pit bulls. Similarly, the last column of Table 1 shows that thinking African American men are violent was the only negative stereotype that significantly predicted rating pit bulls less favorably than other dog breeds. Those results both confirm H3's expectations and speak to the importance of examining the racialization process through intersectional analyses of both race and gender schemas.

In sum, anti-Black attitudes are a remarkably consistent predictor of public opinion about pit bulls. Combining the results across twenty models from two different datasets in Table 1, Tables A2, and A3 in S1 Appendix shows that three blatant measures of racial prejudice—anti-Black affect, old fashioned racism, and negative stereotypes of Black men—are all significant predictors of harboring unfavorable opinions of the breed. Those results should not be susceptible to the same endogeneity issues as analyses of how racial attitudes predict public opinion about affirmative action, welfare, and Donald Trump [69,75,76], either, since pit bulls probably aren't salient enough to alter feelings towards African Americans. Nor is omitted variable bias a threat to the validity of our findings. Aside from age, including additional factors (e.g. income, region, homeownership etc.) in our analyses never weakened the results. You can see in the tables, in fact, that normally potent sociopolitical and socioeconomic predictors of public opinion (e.g. partisanship and education) are unrelated to views of pit bulls. The findings, instead, indicate that pit-oriented opinions have much more to do with racial and generational dynamics than partisan and economic polarization.

## Implicit racialization and public opinion

The results presented thus far suggest that the racialization of pit bulls has eroded public support for the breed. So does the fact that the majority of Americans who think of pit bulls as Black-owned dogs harbor more negative opinions of the breed. In our Lucid surveys, for example, 60 percent of respondents who thought white people were more likely to own pit bulls rated the breed favorably, compared to 47 percent of those who think of pit bulls as a Black-owned breed.

Those findings fit a familiar pattern. We noted earlier that overestimating the share of violent crime committed by African Americans is associated with support for more punitive criminal justice polices [21]. Likewise, our analysis of a November 2017 Poll conducted by Survey Sampling International (raw data accessed from the Roper Center), found that 44% of whites who said "poor Black people are more likely to benefit from welfare programs than poor white people" wanted funding for welfare programs to be decreased, compared to just

## "It should be legal to own a Pit Bull"

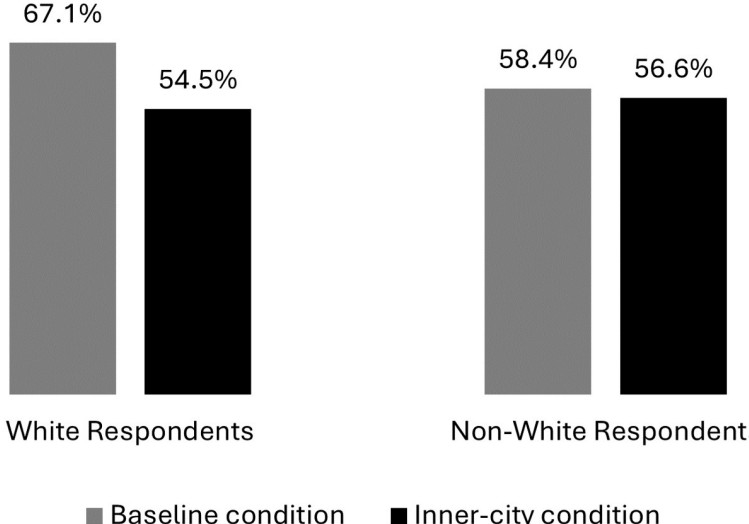

**Fig 4. Opinions of pit bull legalization by race and experimental condition.** The baseline condition asked, "Do you think that it should be legal or illegal to own a pit bull?" The inner-city condition asked, "Do you think that it should be legal or illegal to own a pit bull in inner-city neighborhoods?" *Source*: Pooled Lucid Surveys, June 2020, July 2020, August 2021.

28% of whites who didn't think that welfare disproportionately benefits African Americans. And only 26% of whites who thought Obama's health care proposals favored Blacks over whites supported his universal health care plan, compared to 61% support among whites who didn't think Obamacare would disproportionately benefit African Americans [25].

But the observational nature of these correlations makes it difficult to establish any causal role of racialization in public opinion. So, we tested H4's causal claim that implicitly associating pit bulls with African Americans further erodes white support for the breed by embedding an experiment into three of the Lucid surveys that we fielded in June 2020, July 2020 and August 2021 (pooled N = 3196). Our experimental design followed several other studies, which all implicitly associated African Americans with specific policies through the racially evocative term "inner-city." The inner-city frame is particularly relevant for our purposes since it closely mirrors the way in which scholars and journalists contend that pit bulls have been racialized in media and popular culture. "Once the pit bull was portrayed as an 'inner-city dog,' Dickey [18, pg 146] for instance stated, "it became a magnet for racial fears about crime and the American underclass." Consistent with that contention, Americans greatly exaggerate the share of Black people living in those urban areas [28], making "inner-city" a powerful race-coded cue. So much so that prior experiments, which use "cities" to implicitly prime race, have altered public opinion about criminal justice policies, education spending, and the minimum wage [2,28,43].

The results in Fig 4 show that implicitly associating African Americans with pit bulls via the "inner city" treatment also significantly decreased white Americans' support for legalizing the breed. The left-hand side of the display shows that 67.1% of white respondents said it should be legal to own pit bulls in the baseline wording, compared to only 54.5% of whites in the inner-city condition. That 12.6 percentage-point difference between conditions was very

highly significant, too (T = 6.3, see Table A5 in S1 Appendix), and therefore supports H4's expectations. The right-hand side of the display, meanwhile, shows that non-whites' positions were statistically equivalent regardless of whether they received the inner-city question wording. In fact, we can be quite confident that whites were more heavily influenced by the inner-city treatment than non-whites (p = .006, see Table A3 in S1 Appendix). The highly significant negative interaction here between inner-city*white dovetails with prior experimental research showing that implicit race associations like inner-city are more potent in public opinion for whites than Black Americans [54]; and it suggests that our intended racialization experiment is tapping into something distinctly racial.

Contrary to some prior research, however, unfavorable views of African Americans were not significantly stronger predictors of support for banning the breed in the inner-city condition than they were in the baseline group. This null effect for the inner-city*Black favorability interaction most likely stems from the fact that racial attitudes were already associated with public opinion about pit bulls to begin with. Experiments generally produce weaker racial priming effects on issues that were already racialized [2,25,52]; and we know from the prior sections that most Americans already think of pit bulls through a racialized prism. Interaction effects are also much more difficult to detect statistically [77], as it takes "16 times the sample size to estimate an interaction than to estimate a main effect" [78]. We're especially encouraged, then, by the highly significant interaction between inner-city*white in our experimental results.

But it could still simply be the case that whites evaluate just about any issue or policy less favorably that's implicitly associated with African Americans via inner-city connotations. A recently published article on "Racialized Names and Time to Adoption in a County Dog Shelter," however, shows that implicit associations with African Americans are particularly powerful for pit bulls. "Pit bulls with increasingly Black-sounding names were adopted significantly slower," the study found, "suggesting that adopters were resistant to dogs with Black-sounding names but only when their breed [of pit bull] made race particularly salient" [79, pg. 227]. We suspect that connecting pit bulls to inner cities makes race similarly salient in eroding white support for these dogs.

Yet while the results are consistent with both that interpretation and prior research on how racialization erodes white support for issues and policies associated with African Americans, it is important to conclude here by noting that we cannot say for certain what the exact mechanism is behind why pit bulls are less popular when they're framed as inner-city dogs. It's possible, for instance, that there's less support for large, muscular dogs in cities because people think they need more space to flourish. So, we hope that future research will build on these results by further examining how and why implicit associations with African Americans erode whites' support for legalizing pit bulls.

## Racial prejudice and policies across the states

We noted earlier that the racialization of public opinion often has important policy consequences; and our final hypothesis suggested that the same might be true for pit bulls. To test that contention, Fig 5 displays the relationship between enacting pit bull protection laws and the two different measures of state-level racial prejudice discussed in the data and methods section. You can see that both of those measures—state-level opposition to interracial dating and relative rates of racist Google searches—powerfully predict differences in breed specific legislation (BSL) across the states. Fig 5 shows that states with low levels of racial prejudice had roughly an even chance of passing BSL preemption laws. But those probabilities sharply decline in both displays as we move across the spectrum towards more racially prejudiced

**Fig 5. State-level racial prejudice negatively predicts which states have passed laws preempting breed-specific legislation.** Note: Analysis limited to continental United States. Lines are smoothed averages (bandwidth = .80) *Source*: Data on state laws from Fix and Mitchell; state-level white opposition to interracial dating from the 1987–2012 Pew Values Survey Merged File; data on racist searches are from Google Trends, 2004–2019.

states like West Virginia and Mississippi. The negative relationships between racial prejudice and BSL preemption laws in the two displays are both statistically significant, as well, even with the small number of cases in the analysis (see Table A6 in S1 Appendix).

Those significant negative relationships remain mostly intact after accounting for other factors. The full results in Table A6 in S1 Appendix, for example, show that controlling for state-level attitudes from the Pew data, such as white partisanship and white support for limited government, neither reduced the substantive importance nor the statistically significant relationships between state-level prejudice and pit bull policies. Including each state's Black population percentage introduces more uncertainty into the small-N analyses because it's highly correlated with both measures of state-level prejudice (r = .57 and r = .58 respectively). Yet even with that multicollinearity, state-level racist Google searches remain a significant negative predictor of passing BSL preemption laws (p = .025); and white opposition to interracial dating just misses the mark (p = .15).

So, while there's necessarily more uncertainty in these small-N state-level analyses than in our large-N survey results, the results generally confirm H5's expectations that racially prejudiced states are less likely to enact legislation protecting pit bulls from local bans on the breed. They also amplify prior research on how policies, such as crime, welfare, and Medicaid, tend to be implemented across the states in ways that reflect their underlying racialization.

## Conclusion

Taken together, the results in this article confirm what many keen observers of the pit bull's position in American society have long suspected: Pit bull prejudice is at least partly an extension of racial prejudice that's activated by the breed's affiliation with Black men and African American culture. While that's certainly significant, the article's more important contribution comes from what these dogs can teach us about racial politics. By providing a novel empirical

application that synthesizes and extends existing research on how the racialization process affects public opinion and policy implementation across the states, our analyses of pit bulls provide broader insights into the ongoing power of anti-Blackness in American politics.

Indeed, we've uncovered some strong similarities between the racial politics surrounding pit bulls and the well-documented racialization of other ostensibly non-racial issues, such as crime and welfare. In each instance, there is some empirical link between African Americans and the issue. African Americans are more likely to be incarcerated, receive welfare, and to interact with pit bulls than white people. But those associations are then framed in an especially negative light by the news media, which in turn erodes racially prejudiced whites' support for welfare, compassionate criminal justice policies, and pit bulls. The implementation of these racialized policies across the states then reflects their underlying racialization, as the most racially prejudiced states have less generous welfare benefits, more punitive crime policies, and fewer pit bull protection laws. To be sure, racialization is far from the only reason why so many people dislike pit bulls—and it may not even be the most important factor. But the results in the article indicate that it's impossible to disentangle pit bull politics from the politics of race.

Well, up until very recently at least. Unlike the enduring associations between African Americans and issues like welfare and crime, the racialization of pit bulls appears to be changing rather rapidly. While the news media's stories about pit bulls have been consistently negative, our related research shows that depictions of pit bulls on social media are tremendously positive [39]. So much so, that there are actually more positive tweets and Instagram posts about pit bulls than there are about any other dog breed. That positive imagery predominantly has a white face attached to it, as pit bull positivity on social media is overwhelmingly propagated by white advocates doting on the breed [10,39].

The changing face of "pit bull people" from Black to white Americans is making the breed more popular, too—much the way that white people have become more supportive of drug treatment now that the face of the opioid epidemic is increasingly white [34]. In fact, our subsequent research shows that public opinion is increasingly shifting in favor of pit bulls; and more significantly, that the racialized (white) pit bull positivity that characterizes so much social media content has disproportionately affected white Americans' opinions of the breed. Like other prominent issues, pit bull negativity is racialized as Black but pit bull positivity has a white face.

## Supporting information

**S1 Appendix.**
(DOCX)

**S1 File.**
(ZIP)

## Author Contributions

**Conceptualization:** Michael Tesler, Mary McThomas.

**Data curation:** Michael Tesler.

**Formal analysis:** Michael Tesler.

**Methodology:** Michael Tesler.

**Writing – original draft:** Mary McThomas.

**Writing – review & editing:** Mary McThomas.

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
