## [Decision Letter · Decision Letter 0]

10 Oct 2023

PONE-D-22-34733The Racialization of Pit Bulls: What Dogs Can Teach Us About Racial PoliticsPLOS ONE

Dear Dr. Tesler,

Thank you for submitting your manuscript to PLOS ONE. After careful consideration, we feel that it has merit but does not fully meet PLOS ONE’s publication criteria as it currently stands. Therefore, we invite you to submit a revised version of the manuscript that addresses the points raised during the review process. As you can see from the reviewers' reports, reviewers were generally positive about the paper but also raised various issues about the methods you used and the way in which data and results were presented. Please revise your manuscript according to their suggestions. 

We look forward to receiving your revised manuscript.

Kind regards,

Hans H. Tung

Academic Editor

PLOS ONE

Journal Requirements:

Additional Editor Comments (if provided):

Note from the PLOS Editorial Office: Please note that reviewer 1 is Justin T Pickett, who has agreed to be named as a reviewer.

Reviewers' comments:

Reviewer's Responses to Questions

**Comments to the Author**

1. Is the manuscript technically sound, and do the data support the conclusions?

Reviewer #1: Yes

Reviewer #2: Partly

Reviewer #3: Partly

2. Has the statistical analysis been performed appropriately and rigorously? 

Reviewer #1: Yes

Reviewer #2: Yes

Reviewer #3: Yes

3. Have the authors made all data underlying the findings in their manuscript fully available?

Reviewer #1: Yes

Reviewer #2: Yes

Reviewer #3: Yes

4. Is the manuscript presented in an intelligible fashion and written in standard English?

Reviewer #1: Yes

Reviewer #2: Yes

Reviewer #3: Yes

5. Review Comments to the Author

Reviewer #1: This is a fantastic study that has one major flaw: The authors have ignored extremely relevant prior research. And that is totally unacceptable. I will give a clear example, if you go to Google scholar and search for "racial attitudes pit bull," the very first study returned (Thompson et al., 2022) is an experiment that did almost exactly what the current study does (but found different results). It developed the theory for why racial attitudes should be related to views about pit bulls and breed-specific legislation, it measured peoples' perceptions of the racial composition of pit bull owners, it tested whether you could prime racial concerns and influence support for breed-specific legislation, and it also examined the relationship between racial attitudes and views about breed-specific legislation.

True, the current submission is a MUCH better study than Thompson et al. (2022). The former used a smaller sample, weaker measures, and focused only on college students. Still, it is wild that the current authors fail to cite or discuss the earlier study. That is not how science is supposed to work and I expect better. The current authors need to cite Thompson et al. (2022), they need to discuss the findings in that study, and they need to explain how their study goes beyond it and offer some explanation for why the findings in it are different than the earlier ones.

Given the current authors' focus on the perceived racial composition (or racial typification) of dog owners (e.g., Figure 2), I would also direct them, when talking about the racialization of crime (pp. 8-11), to the extensive work on how racial typification of crime (i.e., the perceived racial composition of criminals) is related to support for harsh policies. The authors should start with Chiricos et al.'s (2004) seminal study and go from there in their literature review. The problem is that authors, for reasons that defy logic, have ignored all the relevant public opinion work by scholars like Ted Chiricos and Frank Cullen. The latter, for example, has several important review articles on racial attitudes and criminal justice attitudes (e.g., Cullen et al., 2021). Science is supposed to be cumulative, and the current authors should act like it.

REFERENCES

Chiricos, Ted, Kelly Welch, and Marc Gertz. 2004. Racial typification of crime and support for punitive measures. Criminology 42: 358-390.

Cullen, Francis T., Leah C. Butler, and Amanda Graham. 2021. Racial attitudes and criminal justice policy. Crime and justice, 50(1), 163–245.

Thompson, Andrew J., Justin T. Pickett, and Jonathan Intravia. 2022. Racial Stereotypes, Extended Criminalization, and Support for Breed-Specific Legislation: Experimental and Observational Evidence. Race and Justice 12: 303-321.

Note from the PLOS Editorial Office: Please note that reviewer 1 is Justin T Pickett, who has agreed to be named as a reviewer.

Reviewer #2: This paper studies the racialization of attitudes regarding pit bulls in the United States. More specifically, the paper proposes that negative attitudes toward pit bulls, and a willingness to ban them or otherwise strictly regulate them, are rooted, at least partly, in implicit associations between the breed and African Americans. Thus, the authors argue, those with negative attitudes toward African Americans as a group are more likely than others to have negative attitudes toward pit bulls. The authors present a wealth of survey, media, experimental and policy-outcome evidence in support of their argument. By and large, this is done extremely well and is generally convincing. That said, there are certain problems that I would like the authors to address before I can recommend publication. I also note a number of minor points for the authors at the end of the review.

Before discussing the problems, I should say first that the authors had me at the survey data: that is, I am fully persuaded by the analysis in the “Racial prejudice and public opinion” section. For one thing, I find it very hard to imagine that the relationship between racial and pit bull attitudes runs in the opposite direction. For another, I cannot think of any important omitted variable that isn’t quite closely related to something the authors already control for in some model. For instance, whites (and so those with lower levels of favorability toward African Americans) probably have more past negative experience with dogs (because they are more likely to own and interact with them). But past negative experience is presumably correlated with dog favorability, which is controlled in models reported in Table 1. Anyway, I think the authors could make more of the (many) reasons why we should take their correlational evidence very seriously.

As regards the problems, first, the analysis of coverage of pit bulls isn’t very convincing. It seems to me that the burden of this analysis is to show that coverage of pit bulls – and perhaps, by extension, wider cultural discourse concerning dogs – associates the breed with African Americans very prominently or regularly. But it’s only the very rare pit bull story that mentions African Americans, about 2%. I don’t know exactly how to calibrate this percentage, but it doesn’t sound very high to me. The authors need to provide some basis for thinking that this percentage indicates that associations between pit bulls and African Americans are sufficiently pervasive in the culture that we should expect views of pit bulls to be racialized.

Relatedly, I don’t follow the authors’ analysis of dog-news sentiment. Figure 3 essentially depicts an interaction, with respect to tone, between the type of dog in a story and whether African Americans are mentioned. It seems to me that the observed interaction is the reverse of what we’d expect, under the authors’ argument. That is, if pit bulls were especially affected by association with African Americans, then I would expect the relative impact of that association on sentiment to be greater for pit bulls than for other dogs. But that’s not the case here: for non-pit bulls, association with African Americans increases the negativity of net sentiment by about 60%; for pit bulls, the comparable effect is about 40%.

My other significant concern involves the experimental evidence. I accept that it supports the authors’ hypothesis (H5), but that hypothesis isn’t actually a very specific test of their argument. And other analysis of the experiment seems to threaten the authors’ interpretation of this effect in terms of racialization. Specifically, the inner-city treatment doesn’t increase the impact of racial attitudes (black favorability) on pit bull attitudes – and this is precisely the effect the authors need to establish, i.e., implicitly associating pit bulls with African Americans increases the weight of racial attitudes in pit bull attitudes. Does the interaction between racial attitudes and the treatment at least reach significance among whites?

I have a number of smaller comments below.

- On p. 11, H4 proposes that “stereotypes of Black men as violent will be particularly potent predictors of rating pit bulls unfavorably”. This wording is somewhat vague, but I presume the authors mean something like the effects of these attitudes are stronger than those for other racial attitudes (group favorability, etc.). The authors conclude (pp. 19-21) that the analysis supports their hypothesis. But I see no test here that compares effects across models or otherwise speaks to the “particular potency” of these stereotypes as a determinant of pit bull attitudes.

- On p. 13, the reference to Figure A2 should be to A1.

- At the top of p. 21, the reference to H5 should be to H4.

Reviewer #3: Overall, I think this paper provides strong evidence of a correlation between attitudes towards African-Americans and towards pitbulls, as well as state levels correlation of behavioral measure of attitudes (online searches) to laws banning pitbulls. There is also media and experimental evidence, but are more equivocal, and its difficult to separate competing explanations.

Main comments:

1)

I don’t think Figure 3 (media) supports the thrust of the argument. Stories about pitbulls mentioning African Americans are more negative – but so are stories about non-pitbulls. This could be straightforward anti-Black racism in the media, with no special link to pitbulls. And pitbull stories are more negative than other dogs, with or without mentioning African-Americans, consistent with pitbulls being (perceived to be) more violent regardless of their owners. Looking at the appendix, stories with Rottweilers are more likely to mention Black people then those with pit bulls! This doesn’t mean the theory is necessarily false given the potential implicit associations etc, but it does need to be acknowledged that this evidence is perhaps suggestive, but not watertight.

2)

Similarly, the experiment says that “unfavorable views of African Americans were not significantly stronger predictors”. I assume this is the result of another model with an interaction to look at heterogeneous effects, although that’s not clear. More importantly, it’s true that experiments trying to prime something that is already salient can have null or small effects. If pitbulls are already highly racialized, then asking about pitbulls will already prime racial attitudes, and adding “inner-city” can’t add much more. However, there is clearly an effect from the inner city treatment! I presume what is happening is that everyone shifts toward banning, not just people with negative racial attitudes. This is a problem for the theory, if people move but not in the direction of their racial attitudes.

Perhaps this is because the results are actually not because of race. Maybe people think its ok to own large and aggressive dogs out in rural areas, where there is lots of space, might be needed to guard animals, etc. They are not good in the city, with crowds of people, small apartments, public transit, etc. I bet you could get the same result asking about bans on large pick-up trucks. Or, perhaps this is about location: racial attitude are confounded with geography, and people are fine with pitbulls “over there”, but don’t want them in their own suburbs.

Regardless, it would be best to up-front about these null results, show it in a figure (so we can see the Cis) and discuss possibilities. Is it because of a non-racial effect, or countervailing confounding? Or measurement issues, due to blunt racial attitude questions (maybe the priming happens among people with weakly negative racial attitudes). Maybe its just power: interaction analyses are extraordinarily power-hunger. Including it in the abstract might be good too. Then we can all better learn from this, and people can do follow-up studies which cite this paper.

3)

Why are only whites included in the surveys starting on p.15? Surely the attitude structure of non-whites is the same: they associate African-Americans with pitbulls, so their views toward them is partly a function of views toward African-Americans. If its specifically anti-black views that matter here, there are certainly Latino and Asian Americans with anti-black views. Plus Figures 1 and 2 include all respondents, and they’re analyzed separately in Figure 5. On the other hand, its possible that there are ceiling effects going on, because of the racial attitude questions used. I think the authors should include all respondents, including non-white ones. Or if they have some strong theoretical reason for excluding them, at least a set of models including them in in the appendix and include a footnote telling the reader the results.

4)

Justify the control variables. There is an awful lot of research arguing partisanship is a product of racial attitudes. I’m not sure what including attitudes toward whites would do to the model, but its not clear what its purpose is. And there is no control for income, which is pretty standard and also correlated with dog ownership. See Lenz and Sahn 2020 on related issues. I think the authors should do a bivariate model, and report if the results are very different or not, and if so then which control variables make the big difference. Or, they could put some significant work into justifying why these control variables are really needed, and others not.

Some smaller issues:

Research on racialization, such as Tesler and others, has emphasized that those with positive (pro-Black) racial attitudes become more positive toward racialized issues. Is that the case here? Similerly, the paper makes various claims, such as “the racialization of pit bulls has eroded public support for the breed”. Has it? Or has it just made some people like them more, and others less.

Its good to report on the discussion of the racial attitude measure, that its less correlated with other social attitudes etc. Its not obvious that this is a good thing, or that it will result in conservative estimates. What we really want is a true measure of racial attitudes. I’m no expert, but I wouldn’t assume attenuation bias, since the error isn’t random just noise, its concentrated at one end of the scale. The authors could look carefully at the methodological research on this. Or, they could just point out the properties of the measures used, and say they aren’t sure exactly what effects this could have.

What are the potential generalizability of these findings? As I’m writing this, I noticed that the UK is going to ban the related breed “American XL Bully”. And after some quick googling, it seems like in other countries including Australia and Canada there are also debate about bans on pitbulls. The authors may not be claiming direct evidence, but given that anti-Black racism and pitbull bans both occur in many countries, it seems worth discussing.

My own experience with lucid is the data is quite low quality, worse than other online platforms. What kind of measure did the authors take? Include that somewhere (unless I missed it). Does excluding low quality respondents change the results?

6. PLOS authors have the option to publish the peer review history of their article (what does this mean?). If published, this will include your full peer review and any attached files.

Reviewer #1: No

Reviewer #2: **Yes: **J. Scott Matthews

Reviewer #3: No

---

## [Author Response · Author response to Decision Letter 0]

22 Nov 2023

Comments to the Editor

Thank you very much for the opportunity to revise and resubmit our manuscript with minor revisions. We have worked hard to address all the reviewers’ critiques and to incorporate their thoughtful suggestions into the revised manuscript. 

More specifically, you’ll see from our memo that we have now engaged with the relevant literature that we regrettably omitted from the first submission, deemphasized the automated sentiment analyses, added some key details about measures and results in accordance with the reviewers’ advice, and further qualified our experimental findings with some new caveats. We hope that these changes have now made the manuscript suitable for publication in PLOS ONE. 

Response to Reviewer 1

Thanks so much for the very helpful suggestions about relevant literature to incorporate into the revised manuscript. We regret these omissions from the prior submission and the revised manuscript now cites these studies in detail, paying particularly close attention to Thomspon et al (2022).

(1)R1 writes: “This is a fantastic study that has one major flaw: The authors have ignored extremely relevant literature prior research.” The reviewer goes on to note that it is “totally unacceptable” to not cite Thompson et al. (2022), as their article “developed a theory for why racial attitudes should be related to views about pit bulls and breed-specific legislation, it measured peoples’ perceptions of the racial composition of pit bull owners, it tested whether you could prime racial concerns and influence support for breed-specific legislation, and it also examined the relationship between racial attitudes and views about breed-specific legislation.” 

RESPONSE: We first want to thank the reviewer for the kind words about our study. Second, we apologize for our failure to cite this important study. Not doing so was an honest omission that simply stemmed from us writing our initial draft of the manuscript long before that article was published. So, we’re very grateful that the reviewer brought these findings to our attention, and we’re pleased to highlight them prominently in our revised manuscript (more on that below). 

(2) R1 relatedly writes, “True, the current submission is a MUCH better study than Thomspon et al. The former used a small sample, weaker measures, and focused only on college students. Still…the current authors need to cite Thompson et al. (2022), they need to discuss the findings in that study, and they need to explain how their study goes beyond it and offer some explanation for why the findings in it are different than the earlier ones.” 

RESPONSE: We, once again, thank the reviewer for the nice words about our study and express sincere regrets over the omission of Thomspon et al. (2022) from our first submission. 

We also really appreciate R1 pushing us to explain why the findings in our study may differ from theirs and now devote 1.5 paragraphs at the front-end of the manuscript to discussing these findings and offering informed explanations for why ours are different. After noting that there have been few empirical analyses testing scholarly and journalistic contentions about the racialization of pit bulls, the revised manuscript now states the following: 

“Moreover, Thompson, Pickett, and Intravia’s (2022) recent empirical study found that neither experimentally linking pit bull owners with African Americans via racial imagery, nor harboring stereotypes of pit bull owners as disproportionately Black, were associated with support for breed-specific legislation among college students.

 Thompson et al (2022), however, conclude that study by noting the need to replicate their findings with representative samples. They specifically discuss how the young age of their college sample could affect the results if attitudes toward pit bulls have improved since Michael Vick’s 2007 dog-fighting controversy. This is a particularly prescient point. Consistent with the unusually low levels of support for breed-specific bans in Thompson et al.’s data (1.74 on a 1-5 scale), we note multiple times throughout the manuscript that age is probably the strongest predictor of public opinion about pit bulls presumably because American youths have been socialized amidst increasingly positive images of the breed on social media. Younger and college-educated Americans score significantly lower in racial prejudice (see discussion below), as well, which could help further explain why the link between pit bulls and African Americans did not increase support for breed-specific legislation among college students.” 

(3) R1 writes, “Given the current authors' focus on the perceived racial composition (or racial typification) of dog owners (e.g., Figure 2), I would also direct them, when talking about the racialization of crime (pp. 8-11), to the extensive work on how the racial typification of crime (i.e., the perceived racial composition of criminals) is related to support for harsh policies. The authors should start with Chiricos et al.'s (2004) seminal study and go from there in their literature review.” 

RESPONSE: We regret the omission and acknowledge our embarrassing blind spot here as political scientists to some of the relevant literature in other disciplines. We are, therefore, grateful to the reviewer for alerting us to the relevant work on this topic in criminology. The Chiricos et al. piece on the racial typification of crime and support for punitive measures was particularly helpful and we now cite that study three different times in our revised manuscript. We also cite additional studies on the racialization of crime that we picked up from R1’s suggestion to look to the lit reviews in Chiricos et al. and Cullen et al. for guidance. 

So, we want to conclude here by sincerely thanking the reviewer again for his help with these important additions to the revised manuscript. 

Response to Reviewer 2

Thank you for both the kind words about our manuscript and for your thoughtful critiques of it. Your concerns about the automated sentiment analyses were particularly important in improving the revised manuscript. 

(1) R2 writes, “I am fully persuaded by the analysis in the “Racial prejudice and public opinion” section. For one thing, I find it very hard to imagine that the relationship between racial and pit bull attitudes runs in the opposite direction. For another, I cannot think of any important omitted variable that isn’t quite closely related to something the authors already control for in some model…I think the authors could make more of the (many) reasons why we should take their correlational evidence very seriously.” 

RESPONSE: We are SO very grateful to the reviewer for these comments. It is indeed difficult to imagine how these results would be susceptible to the same endogeneity issues found in the correlations between racial attitudes and support for affirmative action (Sniderman and Carmines 1997), welfare (Goren 2022), and Trump (Griffin et al. 2021). Nor is omitted variable bias a threat to the statistical significance of the findings. The relationship between racial attitudes and public opinion remained intact in the face of every model specification we tried, as there were only three consistent predictors of public opinion about pit bulls across the hundreds of regressions we’ve run--age, racial attitudes, and general favorability towards dogs--all of which are include in our multivariate analyses. 

We were quite happy, then, to heed the reviewer’s advice and make more out of why we should take “the correlational evidence very seriously.” The concluding paragraph of this section in the revised manuscript now offers reasons why we’re confident that these relationships are not the product of endogeneity or omitted variable bias. The new discussion of omitted variable bias and the robustness of the findings across every model imaginable also indirectly addresses some of the concerns about model specifications raised by R3 (see below) 

(2) R2 writes, “the analysis of coverage of pit bulls isn’t very convincing. It seems to me that the burden of this analysis is to show that coverage of pit bulls – and perhaps, by extension, wider cultural discourse concerning dogs – associates the breed with African Americans very prominently or regularly. But it’s only the very rare pit bull story that mentions African Americans, about 2%. I don’t know exactly how to calibrate this percentage, but it doesn’t sound very high to me. The authors need to provide some basis for thinking that this percentage indicates that associations between pit bulls and African Americans are sufficiently pervasive in the culture that we should expect views of pit bulls to be racialized.”

(3) R2 relatedly writes, “I don’t follow the authors’ analysis of dog-news sentiment. Figure 3 essentially depicts an interaction, with respect to tone, between the type of dog in a story and whether African Americans are mentioned. It seems to me that the observed interaction is the reverse of what we’d expect, under the authors’ argument. That is, if pit bulls were especially affected by association with African Americans, then I would expect the relative impact of that association on sentiment to be greater for pit bulls than for other dogs. But that’s not the case here: for non-pit bulls, association with African Americans increases the negativity of net sentiment by about 60%; for pit bulls, the comparable effect is about 40%.”

RESPONSE: These two passages above are both very fair critiques and it’s now clear to us from the reviews provided by R2 and R3 that the revised manuscript needed to overhaul how we present our automated content analyses. 

To be sure, we still think the analyses in Figure 3 of the original manuscript are an important part of the story since racialized stories about pits are 20-points more negative than those that don’t explicitly reference African Americans and that there’s a larger total volume of negative racialized news stories about pit bulls than all the other dogs we looked at combined. We also think these numbers here are an enormous underestimation of negative racialized news coverage, as qualitative analyses suggest that negative associations between African Americans and pit bulls in media and pop culture occur largely through imagery--and our automated content analyses can only identify explicit references to race. 

But we also completely agree with R2 and R3 that our automated content analyses just aren’t as convincing as the other quantitative analyses, and the evidence in support of H1 is not as strong as the data testing H2-H6 in the original manuscript. So, after careful consideration we decided to jettison the “Negative News Coverage” section from the revised manuscript. 

Instead, the revision makes only brief reference to our “suggestive quantitative evidence” from Crimson Hexagon in the “Theoretical Background and Empirical Expectations” section. After discussing prior qualitative accounts of how media and pop-culture portraits negatively depict pit bulls when they’re linked up with African Americans in more detail than the original submission, the revised manuscript goes on to state:

"And our subsequent research provides some suggestive quantitative evidence on the heightened negativity of news coverage when pit bulls are associated with African Americans (McThomas and Tesler 2024). Drawing on automated sentiment analyses from Gary King’s analytics platform, Crimson Hexagon, we found that the net sentiment (positive minus negative) of headline news stories about pit bulls that explicitly mentioned African Americans was significantly lower (p<.001) than the already high levels of negativity in all news coverage of the breed (-68 to -48 respectively)." 

We think that restructuring our discussion of media portraits this way has several benefits. First, and most importantly, deemphasizing the media findings leaves the manuscript less vulnerable to criticism since that was our weakest hypothesis test. Second, the media findings served largely to bolster empirical expectations about the link between racial attitudes and public opinion. Little is lost, then, from briefly referencing these automated content analyses in the theoretical background and empirical expectation section instead of dedicating a whole empirical section to them. Indeed, almost none of the article-length studies cited on how crime, welfare, and drug addiction have been racialized in public opinion via media portraits include original content analyses of news coverage. Third, referring the reader to the more detailed presentation of the automated content analyses from our book manuscript in progress, which includes an entire chapter on race and media portraits of pit bulls, allows us to address the critiques raised by the reviewers in much greater detail than we could do in the space allotted to a single section of an article. Finally, removing this section provides us with additional space to incorporate the reviewers’ other suggestions. The revised manuscript’s text (minus references) is nearly the exact same length as the original. 

We are, therefore, most grateful to R2 and R3 for convincing us to do something we had been stubbornly reluctant to do even after receiving similar feedback from our colleagues. Upon closer reflection, we think there is a significant addition-by-subtraction effect here and that the manuscript is considerably strengthened by removing this material. 

(4) R2 writes, “My other significant concern involves the experimental evidence. I accept that it supports the authors’ hypothesis (H5), but that hypothesis isn’t actually a very specific test of their argument. And other analysis of the experiment seems to threaten the authors’ interpretation of this effect in terms of racialization. Specifically, the inner-city treatment doesn’t increase the impact of racial attitudes (black favorability) on pit bull attitudes – and this is precisely the effect the authors need to establish, i.e., implicitly associating pit bulls with African Americans increases the weight of racial attitudes in pit bull attitudes.”

RESPONSE We appreciate this valid critique, especially since it also dovetails with some of R3’s comments below. The revised manuscript, therefore, takes multiple steps to address it. 

First, we provide more reasons for why we do not think that the null interactive effect here for inner-city*Black favorability is a serious threat to interpreting the experiment in terms of racialization. In addition to our discussion of the prior studies showing that it’s a lot more difficult to experimentally prime racial attitudes in issues that are already racialized like pit bulls, we’ve incorporated R3’s comment into the revised manuscript about how interaction analyses are “extraordinarily power-hungry.” We now cite Andrew Gelman’s analysis, “You Need 16 Times the Size to Estimate an Interaction Than to Estimate a Main Effect” among others in that discussion and then use it as a segue to further highlight the significant interactive effect for inner-city*White. We think that significant interaction greatly aids the interpretation in terms of racialization, as it suggests that the inner-city treatment is tapping into something distinctly racial; we now explicitly say so in the revised manuscript. 

At the same time, though, we’ve also added some additional caveats to conclude the experimental section, which reflect the legitimate concerns raised by both R2 and R3 (see more below). We think those qualifications help insulate the analysis a bit from critiques and are therefore grateful to both reviewers for their comments on this section. 

(5) R2 includes some more minor comments, as well, such as our misnumbering of hypotheses and figures; the review also notes that the wording of H4 in the original manuscript (“stereotypes of Black men as violent will be particularly potent predictors of rating pit bulls unfavorable”) is vague with regards to what is meant by “particularly potent.”

RESPONSE: We thank the reviewer for identifying these smaller issues. The revised

---

## [Decision Letter · Decision Letter 1]

2 Apr 2024

PONE-D-22-34733R1The Racialization of Pit Bulls: What Dogs Can Teach Us About Racial PoliticsPLOS ONE

Dear Dr. Tesler,

Thank you for submitting your manuscript to PLOS ONE. After careful consideration, we feel that it has merit but does not fully meet PLOS ONE’s publication criteria as it currently stands. Therefore, we invite you to submit a revised version of the manuscript that addresses the points raised during the review process.Many thanks to your patience. We have been trying to collect enough quality reviews for reaching a reasonable decision about your manuscript. While most reviewers have decided to accept the revised manuscript as it stands now, one of them actually made a fairly critical comment on the necessity for this paper to have a more extensive section on data and methods. Despite the other strengths of the current version of the manuscript, we still find it imperative for the paper to fully address the concern raised by this reviewer (#4).  

Please submit your revised manuscript by May 17 2024 11:59PM. We will then quickly make our final decision.  If you will need more time than this to complete your revisions, please reply to this message or contact the journal office at plosone@plos.org. Please include the following items when submitting your revised manuscript:A rebuttal letter that responds to each point raised by the academic editor and reviewer(s). You should upload this letter as a separate file labeled 'Response to Reviewers'.A marked-up copy of your manuscript that highlights changes made to the original version. You should upload this as a separate file labeled 'Revised Manuscript with Track Changes'.An unmarked version of your revised paper without tracked changes. You should upload this as a separate file labeled 'Manuscript'.If applicable, we recommend that you deposit your laboratory protocols in protocols.io to enhance the reproducibility of your results. Protocols.io assigns your protocol its own identifier (DOI) so that it can be cited independently in the future. For instructions see: https://journals.plos.org/plosone/s/submission-guidelines#loc-laboratory-protocols. Additionally, PLOS ONE offers an option for publishing peer-reviewed Lab Protocol articles, which describe protocols hosted on protocols.io. Read more information on sharing protocols at https://plos.org/protocols?utm_medium=editorial-email&utm_source=authorletters&utm_campaign=protocols.

We look forward to receiving your revised manuscript.

Kind regards,

Hans H. Tung

Academic Editor

PLOS ONE

Journal Requirements:

Reviewers' comments:

Reviewer's Responses to Questions

**Comments to the Author**

1. If the authors have adequately addressed your comments raised in a previous round of review and you feel that this manuscript is now acceptable for publication, you may indicate that here to bypass the “Comments to the Author” section, enter your conflict of interest statement in the “Confidential to Editor” section, and submit your "Accept" recommendation.

Reviewer #1: All comments have been addressed

Reviewer #2: All comments have been addressed

Reviewer #4: (No Response)

Reviewer #5: (No Response)

Reviewer #6: (No Response)

2. Is the manuscript technically sound, and do the data support the conclusions?

Reviewer #1: Yes

Reviewer #2: Yes

Reviewer #4: No

Reviewer #5: Yes

Reviewer #6: Partly

3. Has the statistical analysis been performed appropriately and rigorously? 

Reviewer #1: Yes

Reviewer #2: Yes

Reviewer #4: No

Reviewer #5: I Don't Know

Reviewer #6: I Don't Know

4. Have the authors made all data underlying the findings in their manuscript fully available?

Reviewer #1: Yes

Reviewer #2: Yes

Reviewer #4: Yes

Reviewer #5: Yes

Reviewer #6: Yes

5. Is the manuscript presented in an intelligible fashion and written in standard English?

Reviewer #1: Yes

Reviewer #2: Yes

Reviewer #4: Yes

Reviewer #5: Yes

Reviewer #6: No

6. Review Comments to the Author

Reviewer #1: The authors did a phenomenal job revising their manuscript. This is an impressive study that makes an important contribution to the literature. It also is now situated well within the relevant interdisciplinary literature.

Reviewer #2: The authors have done a great job of responding to my concerns and those of the other reviewers. As a result, an already very strong paper is just that much stronger. I have no further revisions or other suggestions to make.

I should note that I understand the authors’ reluctance to exclude the media analysis from the paper, and I agree that the large negative effect of racializing content on the tone of pit bull stories is indicative. I’m glad to hear this content will find a home in the book manuscript.

Reviewer #4: The investigators may have engaged with the relevant literature. However, the synthesis and statistical presentation of the information is awkward and, in places, difficult to follow. For example, in the abstract they list 5 points which they believe they demonstrate. However, not until later in the manuscript do they start addressing what appears to be the five hypothesis, H1 to H5, which this reader assumes are the major goals of the paper. They should have been formerly stated as hypotheses or at least the opposite or alternatives of five null hypotheses at the start of a formal statistical analysis plan. The paper is primarily a descriptive conclusion by the investigators after literature or survey gathering. There is no description of a systematic review of the information gathered and formal planned statistical synthesis of the information. The authors merely pull information from their gathering and then reach the conclusion of a racial bias. One possibility would be a good statistical meta-analysis of the information which would apply to this data with a rigorous statistical conclusion. Also a meta regression would probably point out any possible causes of heterogeneity in the results. The presentation, as it exists, is fragmented.

In addition, there appears to be some relevant information. For example Figure 1 is presented with percentages which is fine. However, as mentioned above, the authors give their take on the results descriptively. In the last bar presentation ‘Rate pit bulls very favorably’ how significant is that two percentage point difference. Probably statistically so with the sample size, but how meaningful practically is that difference?

On page 8 below Figure 2, the authors note that, “It’s hardly a coincidence, either, that the two dog breeds stereotyped as Black-owned are the two breeds that evoke the most fear by far from the public (Tesler 2020).” Has fear and type of breed been statistically associated in the reference? That p-value should be noted. Also throughout the manuscript the words, ‘suggest’ and ‘implicit’, appear which gives the impression of possible author bias, which I’m sure is not the case. See page 22,” The results in Figure 4 show that implicitly associating African Americans with pit bulls via the “inner city” treatment also significantly decreased white Americans’ support for legalizing the breed.” See page 22 to 23, “ The highly significant negative interaction here between inner city white dovetails with prior experimental research showing that implicit race associations like inner-city are more potent in public opinion for whites than Black Americans (White 2007); and it suggests that our intended racialization experiment is tapping into something distinctly racial.” These are not strong statistical arguments.

Some clarity is needed throughout. On Figure 3 what is the reason for some of the interaction across these two surveys. On Table A3 “Old fashion Racism” is a variable. How strongly does that really associate with choosing a dog? There are no measures of concordance given with the logistic presentations. So the strength of the association is not clear. Like wise on Table 1, where are the R-square or partial r-square measures? This brings up a point brought up earlier of missing information. Would it be helpful to have Black American opinions for the information on Table 1?

The paper needs a thorough rewrite showing what can be said definitively and statistically and not necessarily implicit or suggested. Also as a previous reviewer pointed out, how representative of the population is this information? Some demographic descriptions of this sample may be helpful.

Reviewer #5: (No Response)

Reviewer #6: The topic is wonderful, creative, and necessary. However, there are several problems. For example, APA style guidelines are not followed, and conventions of standard American academic English are only erratically shown (e.g., lots of writing in second-person voice; use of contractions; basic grammar errors, such as problems with capitalization).

I would recommend that you talk with a skilled editor who is familiar with both grammar and the conventions of APA style, 7th edition. The writing style is disjointed and difficult to read; the connections among subtopics seems loose at best. Again, while the topic is of critical importance, literature needs to be better used so that it can support the good points that the authors are otherwise trying to make.

I would be happy to elaborate further in a conversation, if that is permitted. There is a lot of room here to create an outstanding, excellent, and transformative article.

7. PLOS authors have the option to publish the peer review history of their article (what does this mean?). If published, this will include your full peer review and any attached files.

Reviewer #1: No

Reviewer #2: **Yes: **J. Scott Matthews

Reviewer #4: No

Reviewer #5: No

Reviewer #6: **Yes: **Adam Duberstein

---

## [Author Response · Author response to Decision Letter 1]

30 Apr 2024

Response to the Editor

Thank you for the opportunity to revise and resubmit the manuscript. We are especially appreciative that you plan to make a quick decision upon receipt of the revised manuscript without another round of reviews. 

We were also particularly pleased to see that R1 and R2 described our response to their critiques as “phenomenal” and “great” respectively, and that R6 described the topic as “wonderful, creative, and necessary.” 

As the memo details, we have restructured the revised manuscript in accordance with your suggestion to have a new section that provides greater detail up front on the data and methods used to test our hypotheses. While we were initially reluctant to make this change, we think it has improved the manuscript and hope that these revisions, combined with the other reviewers’ evaluations, have now made the article suitable for publication in PLOS ONE. 

Response to Reviewer 4

We thank the reviewer for pushing us to provide a more detailed standalone section on the data and methods used in our manuscript. Doing so helped strengthen the revised article. 

(1) R4 first writes, “in the abstract they list 5 points which they believe they demonstrate. However, not until later in the manuscript do they start addressing what appears to be the five hypothesis, H1 to H5, which this reader assumes are the major goals of the paper. They should have been formerly stated as hypotheses or at least the opposite or alternatives of five null hypotheses at the start of a formal statistical analysis plan.”

While our five theoretically informed hypotheses were all formally stated in the “Theoretical Background and Empirical Expectations” section of the prior manuscript, it is certainly reasonable for the reviewer to ask for a more detailed standalone section on the data and methods used to test those expectations. 

The revised manuscript, therefore, now includes a new “Data and Methods” section following the section on “Theoretical Background and Empirical Expectations.” This new section details the various data, measures, and statistical analyses used to test our formal hypotheses. We think that these revisions have improved the manuscript and thank both the reviewer and the editor for pushing us to make them. 

(2) In addition, R4 writes, “there appears to be some relevant information [missing]. For example Figure 1 is presented with percentages which is fine. However, as mentioned above, the authors give their take on the results descriptively. In the last bar presentation ‘Rate pit bulls very favorably’ how significant is that two percentage point difference. Probably statistically so with the sample size, but how meaningful practically is that difference?”

We’d like to respectfully note here that the statistical and substantive significance of this difference were addressed on pg 6-7 of the prior manuscript where we write: "There were not significant differences between the races in how favorably they rated pit bulls; but after controlling for the fact that white people have more favorable views about dogs than Black people do, and that attitudes about dogs in general strongly predict attitudes towards pit bulls (see Table 1), African Americans were significantly more likely than whites to rate pit bulls very favorably (40% to 31% respectively)." 

(3) On page 8 below Figure 2, R4 writes, “the authors note that, ‘It’s hardly a coincidence, either, that the two dog breeds stereotyped as Black-owned are the two breeds that evoke the most fear by far from the public (Tesler 2020).” Has fear and type of breed been statistically associated in the reference?’

The reviewer asks a very fair question here. So, the revised manuscript now elaborates on this point by adding the following information on pg. 8 about the link between fear and breed provided in this reference: 

“Over 40 percent of respondents in a 2018 Lucid survey we conducted said that “scary” described pit bulls and rottweilers ‘extremely’ or ‘very well’ (46 and 41 percent, respectively). But only about 10 percent said the same thing about golden retrievers, collies, Dalmatians and Labradors. These four breeds were all rated at least 15 percentage points more favorably than rottweilers and pit bulls in our surveys as well (Tesler 2020).”

(4) R4 further writes, "Also throughout the manuscript the words, ‘suggest’ and ‘implicit’ appear, which gives the impression of possible author bias, which I’m sure is not the case. See page 22, “The results in Figure 4 show that implicitly associating African Americans with pit bulls via the “inner city” treatment also significantly decreased white Americans’ support for legalizing the breed.”

We were a bit perplexed by this comment, as our repeated use of “implicit” simply refers to the long line of social science literature cited in the article on the impact of messages that indirectly associate a racial group with a policy/person/dog with racially evocative words (e.g. inner-city) without making an explicit reference to that group by name. 

To avoid any potential confusions about the meaning of “implicit” here, we now make further note of this distinction between implicit vs. explicit racial references on pg 13, writing, “Several experiments, which randomly assigned subjects to receive implicit racialized messages (e.g., racial images and/or race-coded language that doesn’t explicitly reference a particular racial group) about specific issues, have affected white Americans’ opinion about crime, welfare, drug treatment, gun control, government spending, education programs, Social Security, the minimum wage, the Iraq War, and the coronavirus pandemic.”

(5) R4 writes, “On Table A3 “Old fashion Racism” is a variable. How strongly does that really associate with choosing a dog? There are no measures of concordance given with the logistic presentations. So the strength of the association is not clear. Like wise [sic] on Table 1, where are the R-square or partial r-square measures?

These are fair questions that we thank the reviewer for raising. The revised manuscript therefore includes R-squared or Pseudo R-squared measures in our regression tables.

---

## [Editor Report · Decision Letter 2]

10 Jun 2024

The Racialization of Pit Bulls: What Dogs Can Teach Us About Racial Politics

PONE-D-22-34733R2

Dear Dr. Tesler,

We’re pleased to inform you that your manuscript has been judged scientifically suitable for publication and will be formally accepted for publication once it meets all outstanding technical requirements.

Kind regards,

Hans H. Tung

Academic Editor

PLOS ONE
---

## [Editor Report · Acceptance letter]

21 Jun 2024

PONE-D-22-34733R2 

PLOS ONE

Dear Dr. Tesler, 

I'm pleased to inform you that your manuscript has been deemed suitable for publication in PLOS ONE. Congratulations! Your manuscript is now being handed over to our production team.

Kind regards, 

on behalf of

Dr. Hans H. Tung 

Academic Editor

PLOS ONE